Breaking the mold: telescoping drives the evolution of more integrated and heterogeneous skulls in cetaceans

Buono Mónica R. buono@cenpat-conicet.gob.ar 1
Vlachos Evangelos 2
1 Instituto Patagónico de Geología y Paleontología, CCT CONICET-CENPAT , Puerto Madryn , Chubut , Argentina
2 CONICET and Museo Paleontológico Egidio Feruglio , Trelew , Chubut , Argentina
Pyenson Nicholas
Electronic publication date: 2022 May 5
Publication date: 2022
Volume: 10
Electronic Location ID: e13392
Received 2021 Nov 4; Accepted 2022 Apr 16
Copyright: ©2022 Buono and Vlachos
Copyright year: 2022
Copyright holder: Buono and Vlachos
License: This is an open access article distributed under the terms of the Creative Commons Attribution License, which permits unrestricted use, distribution, reproduction and adaptation in any medium and for any purpose provided that it is properly attributed. For attribution, the original author(s), title, publication source (PeerJ) and either DOI or URL of the article must be cited.
License URL: https://creativecommons.org/licenses/by/4.0/

Keywords: Anatomical networks, Neoceti, Skull morphology, Modularity, Telescoped bones

Funding: Agencia Nacional de Promoción Agencia Nacional de Promoción de la Investigación el Desarrollo Tecnológico y la Innovación PICT 2019-00327 This works was supported by Agencia Nacional de Promoción Agencia Nacional de Promoción de la Investigación, el Desarrollo Tecnológico y la Innovación (PICT 2019-00327). The funders had no role in study design, data collection and analysis, decision to publish, or preparation of the manuscript.

==============================
Background

Along with the transition to the aquatic environment, cetaceans experienced profound changes in their skeletal anatomy, especially in the skull, including the posterodorsal migration of the external bony nares, the reorganization of skull bones (= telescoping) and the development of an extreme cranial asymmetry (in odontocetes). Telescoping represents an important anatomical shift in the topological organization of cranial bones and their sutural contacts; however, the impact of these changes in the connectivity pattern and integration of the skull has never been addressed.

Methods

Here, we apply the novel framework provided by the Anatomical Network Analysis to quantify the organization and integration of cetacean skulls, and the impact of the telescoping process in the connectivity pattern of the skull. We built anatomical networks for 21 cetacean skulls (three stem cetaceans, three extinct and 10 extant mysticetes, and three extinct and two extant odontocetes) and estimated network parameters related to their anatomical integration, complexity, heterogeneity, and modularity. This dataset was analyzed in the context of a broader tetrapod skull sample as well (43 species of 13 taxonomic groups).

Results

The skulls of crown cetaceans (Neoceti) occupy a new tetrapod skull morphospace, with better integrated, more heterogeneous and simpler skulls in comparison to other tetrapods. Telescoping adds connections and improves the integration of those bones involved in the telescoping process (e.g., maxilla, supraoccipital) as well as other ones (e.g., vomer) not directly affected by telescoping. Other underlying evolutionary processes (such as basicranial specializations linked with hearing/breathing adaptations) could also be responsible for the changes in the connectivity and integration of palatal bones. We also find prograde telescoped skulls of mysticetes distinct from odontocetes by an increased heterogeneity and modularity, whereas retrograde telescoped skulls of odontocetes are characterized by higher complexity. In mysticetes, as expected, the supraoccipital gains importance and centrality in comparison to odontocetes, increasing the heterogeneity of the skull network. In odontocetes, an increase in the number of connections and complexity is probably linked with the dominant movement of paired bones, such as the maxilla, in retrograde telescoping. Crown mysticetes (Eubalaena, Caperea, Piscobalaena, and Balaenoptera)are distinguished by having more integrated skulls in comparison to stem mysticetes (Aetiocetus and Yamatocetus), whereas crown odontocetes (Waipatia, Notocetus, Physeter, and Tursiops) have more complex skulls than stem forms (Albertocetus). Telescoping along with feeding, hearing and echolocation specializations could have driven the evolution of the different connectivity patterns of living lineages.

Introduction

The skull of crown or living cetaceans (= Neoceti) experienced dramatic changes throughout its evolutionary history, related to the rearrangement of cranial bones and the acquisition of a novel feature in mammalian skull configuration, i.e., an extreme telescoping. Telescoping is defined as a combination of extensive bone overlap and extreme proximity of occipital-rostral elements (Miller, 1923; Roston & Roth, 2019) which distinguishes crown cetaceans from the stem forms (informally known as “archaeocetes” and defined by the retention of plesiomorphic features). Cetacean telescoping also promotes changes in the connections between bones and new types of cranial sutures (= horizontal sutures; Gatesy et al., 2013; Roston & Roth, 2019). This represents an emerging level of bone-suture configurations, breaking the typical mammalian skull design and providing new morphospaces which might facilitate the exploration of new ecological and behavioural strategies. Within neocetes, two types of telescoping are recognized, one in each group of living cetaceans (Fig. 1): one dominated by the posterior expansion of anterior bones (= retrograde cranial telescoping sensu Churchill et al., 2018) typical of odontocetes or toothed whales, and the other dominated by forwarding movement of posterior bones (= prograde cranial telescoping sensu Churchill et al., 2018) found in mysticetes or baleen whales (Miller, 1923; Kellogg, 1928a; Kellogg, 1928b).

Figure 1 The two different types of telescoping in neocetes, from skull to network.

Although externally the connectivity pattern of the two telescoping types is similar (see the simplified drawings), internal bones are connected in a very different way. In the prograde telescoping seen in mysticetes (here represented by Balaenoptera spp.) additional connections are seen in the supraoccipital and the ventro-lateral parts of the skull. In the retrograde telescoping seen in the odontocetes (illustrated by Tursiops), numerous new connections are modeled in the internal (e.g., vomer) and ventral parts of the skull (corresponding to the palatal region).

A recent morphometric analysis of the skull of odontocetes suggested three phases in the evolution of facial morphology and cranial telescoping (Churchill et al., 2018): the first phase, in which the lateral expansion of the maxilla is limited and the intertemporal region is broadly dorsally exposed, and premaxilla, nasal and external bony nares are anterior to the orbits (typical of Xenorophidae and Simocetus, among other stem forms); the second phase, in which a further posterior displacement of the nares and surrounding bones (nasal, premaxilla and maxilla) is observed and the intertemporal region is not visible in dorsal view any more (this condition is described for waipatiids and squalodontids); and the final phase characterized by an increased overlap of the frontal and maxilla (observed in crown odontocetes). Among mysticetes or baleen whales, different types of telescoping are described by Miller (1923; P:20-22), characterizing the main families of baleen whales; however, quantitative analyses as those performed by Churchill et al. (2018) in odontocetes (which include a small sample of mysticetes) are still required.

While telescoping has been investigated in the last few years using different methodologies and approaches (e.g., Churchill et al., 2018; Roston & Roth, 2019), the impact of the novel suture configurations in the topographical organization and integration of the cetacean skull has never been addressed. Anatomical Network Analysis (AnNA) has recently emerged as a new tool to quantify the complexity of anatomical structures as a function of their pattern of organization, in which bones, suture joints, and contacts between the bones are modeled as the nodes and links of a network (Rasskin-Gutman & Esteve-Altava, 2014). This methodology allows the study of a level of morphological information that has been seldom analyzed, the level of connections, complementing an integral morphological approach (Esteve-Altava, 2013). The solid theoretical foundations of the AnNA (Esteve-Altava, 2013; Rasskin-Gutman & Esteve-Altava, 2014) has allowed its successful application in various anatomical structures, like the mammalian skeleton (Powell et al., 2018), tetrapod skull (e.g., Esteve-Altava et al., 2013a; Esteve-Altava et al., 2013b; Esteve-Altava & Rasskin-Gutman, 2014; Lee, Esteve-Altava & Abzhanov, 2020) and tetrapod limbs (Molnar et al., 2017; Esteve-Altava et al., 2018; Esteve-Altava et al., 2019; Fernández et al., 2020), among many other studies. In particular, a recent AnNA analysis of 44 tetrapod skulls by Esteve-Altava et al. (2013a) revealed that the reduction in the number of skull bones during tetrapod evolution increased the complexity of the connectivity pattern under a regime of important structural constraints.

In this study, we expand the Esteve-Altava et al. (2013a) sample with the addition of stem cetaceans, odontocete and mysticete skulls, and apply AnNA analysis to this sample to first examine if the connectivity pattern of the cetacean skull is the same as that observed in other mammals or if the changes in the topological organization of the bones produced by telescoping also affected the connectivity pattern of the crown cetacean skulls. In addition, we want to test if the connectivity pattern between prograde and retrograde telescoping of the cetacean skull is different overall and at an individual bone level. We anticipate that the results of this work will add meaningful interpretations applicable to ongoing neontological and paleontological discussions of cetacean evolution.

Materials & Methods

Sample

Considering that the main goal of this study is to identify the connectivity pattern of the cetacean skull and its relation with the evolution of telescoping, we gathered the sample that encompasses the main patterns of telescoping identified in major clades (see more details in Churchill et al., 2018). We constructed networks for 21 cetacean skulls covering the main lineages of cetaceans: three stem cetaceans (Pakicetus, “Protocetidae”, and Dorudon), three extinct (Aetiocetus, Yamatocetus, and Piscobalaena) and ten extant mysticetes from three genera (Eubalaena, Caperea, and Balaenoptera spp.), three extinct (Albertocetus, Waipatia, and Notocetus) and two extant odontocetes (Physeter and Tursiops). In the case of Balaenoptera we included all the extant species known to test if the interspecific variation observed in the skull vertex of these species impacts the pattern of connectivity (see Table S1 for more details of the sample). For Pakicetus and “Protocetidae” (a paraphyletic group; see for example Gohar et al., 2021), due to the lack of complete fossil skulls, we constructed an average skull network for each taxon based on different species. In protocetids, this model was based on Aegyptocetus, Georgiacetus, a “Protocetidae” indet. and Artiocetus, while for Pakicetus was based on the most complete specimens of P. attocki and P. inachus (Table S1). In the case of Tursiops—the only cetacean included in the Esteve-Altava et al. (2013a) dataset—the anatomical network is new and based on our own observations of T. truncatus.

Construction of the networks

The networks were constructed manually based on photographs, drawings, descriptions, and/or first-hand examinations of the specimens (Table S1 and Data S1–S2), considering the bones as the nodes and suture joints/bone contacts as links of the network. Bone contacts/sutures were determined based on observations of adult specimens. However, to check internal sutures and/or contacts of completely fused bones we include juvenile specimens in the sample. In those cases in which bone contacts could not be determined, we either assumed the same condition as observed in closely related taxa (if available) or did not model them. The interparietal, a bone that is difficult to trace in most cetacean skulls, was no modeled in stem cetaceans because its presence has not been reported in any of the taxa analyzed. In neocetes, it was modeled as fused to the supraoccipital (see Mead & Fordyce, 2009) unless evidence of its dorsal exposition was found in the vertex of adult specimens (in which case we modeled all the observed contacts). For odontocetes, any evidence of asymmetry in the skull (see Coombs et al., 2020) expressed in the connectivity pattern has been taken into account.

Even though the telescoped sutures of cetaceans present a different pattern in comparison to other mammals (Roston & Roth, 2019), they were modeled in the anatomical networks as links with the same weight.

All the anatomical networks were digitized in Gephi (Bastian, Heymann & Jacomy, 2009).

Anatomical network analysis

The cetacean networks were included in the dataset of Esteve-Altava and collaborators (2013a), expanding the tetrapod sample. Five main descriptors were used to characterize the networks: Density (D, the complexity of the anatomical structure, calculated as the ratio between the connections and the maximum possible connections), Heterogeneity (H; the differentiation of the connections of the various nodes, calculated as the ratio between the standard deviation of the connections along the network and the average number of connections), Average Clustering Coefficient (C; the anatomical integration of the various nodes with their surroundings, calculated as the average of the clustering coefficient of each node that measures the connections between the neighbors of each node), Parcellation (P, the degree of anatomical modularity of the network, based on the distribution of nodes in the different recovered modules); Average Path Length (L; the anatomical integration of the various nodes related to their effective proximity, calculated as the average number of steps between any pair of nodes), based on Esteve-Altava et al. (2013a), Esteve-Altava et al. (2013b), Esteve-Altava et al., 2014; Esteve-Altava et al. (2018), Esteve-Altava et al. (2019), and Esteve-Altava & Rasskin-Gutman (2014); Parcellation was calculated as in Fernández et al. (2020). Our data are summarized in Table 1; see supplemental information as well. For the analysis at the individual bone level, we used the Clustering Coefficient of each bone (CluC), and the three main centrality measures: the Degree Centrality (DeC, how many connections a node has), Closeness Centrality (CloC, the average of the shortest path of a node with any other node of the network), and Betweenness Centrality (BetC, how many times a node is included in the shortest path of any other pair of nodes) (Esteve-Altava, 2013). These metrics have been used to calculate the various graphs; Principal Component Analysis (PCA) under correlation (normalized var-covar) and PERMANOVA were performed in PAST v. 4.0 (Hammer, Harper & Ryan, 2001).

Table 1 Main descriptors of the cetacean sample analyzed.

Taxon	Group	Category	N	K	D	L	C	H	P	UBR	
Pakicetus	Stem Cetacea	Extinct	35	99	0.166	2.334	0.391	0.404	0.731	0.2	
“Protocetidae”	Stem Cetacea	Extinct	35	102	0.171	2.366	0.540	0.435	0.744	0.2	
Dorudon	Stem Cetacea	Extinct	35	92	0.155	2.361	0.439	0.542	0.741	0.2	
Aetiocetus	Mysticeti	Extinct	35	78	0.131	2.533	0.487	0.563	0.793	0.2	
Yamatocetus	Mysticeti	Extinct	35	82	0.138	2.432	0.415	0.589	0.748	0.17	
Piscobalaena	Mysticeti	Extinct	35	85	0.143	2.450	0.478	0.533	0.738	0.17	
Caperea	Mysticeti	Extant	35	87	0.146	2.378	0.465	0.568	0.795	0.2	
Eubalaena	Mysticeti	Extant	35	95	0.160	2.333	0.489	0.581	0.795	0.2	
Balaenoptera acutorostrata	Mysticeti	Extant	35	89	0.150	2.407	0.511	0.531	0.738	0.2	
Balaenoptera borealis	Mysticeti	Extant	35	87	0.150	2.434	0.509	0.514	0.738	0.2	
Balaenoptera edeni	Mysticeti	Extant	35	89	0.150	2.407	0.511	0.531	0.738	0.2	
Balaenoptera brydei	Mysticeti	Extant	35	89	0.150	2.407	0.511	0.531	0.738	0.2	
Balaenoptera musculus	Mysticeti	Extant	35	91	0.153	2.324	0.512	0.565	0.797	0.2	
Balaenoptera omurai	Mysticeti	Extant	35	85	0.143	2.450	0.510	0.549	0.743	0.2	
Balaeonoptera physalus	Mysticeti	Extant	35	87	0.146	2.440	0.520	0.530	0.784	0.2	
Balaeonoptera ricei	Mysticeti	Extant	35	83	0.139	2.489	0.487	0.529	0.797	0.2	
Albertocetus	Odontoceti	Extinct	35	88	0.148	2.536	0.476	0.476	0.761	0.2	
Waipatia	Odontoceti	Extinct	35	97	0.163	2.457	0.392	0.557	0.669	0.2	
Notocetus	Odontoceti	Extinct	35	105	0.176	2.363	0.450	0.446	0.730	0.2	
Tursiops	Odontoceti	Extant	35	93	0.156	2.506	0.400	0.57	0.793	0.2	
Physeter	Odontoceti	Extant	34	103	0.184	2.228	0.491	0.525	0.666	0.2	
Notes.

C Average Clustering Coefficient

D Density

H Heterogeneity

K Connections

L Average Path Length

N nodes

P Parcellation

UBR unpaired bone ratio

The integration of the networks is defined by the Average Path Length and Clustering Coefficient while the complexity is mainly defined by the Density.

Phylogeny

For the analysis of network descriptors within a phylogenetic context, we constructed a composite phylogeny following Martínez-Cáceres, Lambert & De Muizon (2017) for stem cetaceans, Marx et al. (2019) for mysticetes and Viglino et al. (2021) and Boessenecker, Ahmed & Geisler (2017) for odontocetes. The network descriptors were optimized and mapped under maximum parsimony in the TNT 1.5 software (Goloboff & Catalano, 2016).

Figure 2 PCA of the skull networks of various tetrapods, based on the initial dataset of Esteve-Altava et al. (2013a) and the cetacean sampling added herein.

The first two PCs permit separating the skulls of tetrapods based on their heterogeneity, integration (based on Clustering and Path Length) and complexity (based on Density). Those placed in the first and fourth quadrants show skulls that are better integrated, further divided into those with more heterogeneous (first quadrant) or more complex (fourth quadrant) skulls. Those placed in the second and third quadrants show skulls that are less integrated, further divided into those with more homogeneous (third quadrant) or simpler (second quadrant) skulls. Most derived cetaceans explore a previously unoccupied region for other non-flying tetrapods, with integrated skulls that are quite heterogeneous. Abbreviations: Ans, Anser; Can, Canis; Car, Carettochelys; Che, Chelodina; Chel, Chelydra; Chi, Chisternon; Cor, Corythosaurus; Cro, Crocodylus; Did, Didelphis; Dim, Dimetrodon; Dipl, Diplometopon; Dro, Dromaeosaurus; Enn, Ennantosaurus; Epi, Epicrionops; Gas, Gastrotheca; Gop, Gopherus; Hem, Hemitheconyx; Ich, Ichthyostega; Igu, Iguana; Jon, Jonkeria; Kay, Kayentachelys; Orn, Ornithorhynchus; Pet, Petrolacosaurus; Pha, Phascolarctos; Pla, Plateosaurus; Pod, Podocnemis; Pro, Procolophon; Prog, Proganochelys; Pte, Pteropus; Pyt, Python; Rha, Rhamphorhynchus; Sal, Salamandra; Sey, Seymouria; Sphe, Sphenodon; Ste, Stegosaurus; Sten, Stenocercus; Tes, Testudo; Thr, Thrinaxodon; Tup, Tupinambis; Var, Varanus; You, Younginia.

Results

Cetacean skull networks within tetrapod morphospace

The first two PCAs explained 89% of the variation of the tetrapod skull (PC1: 68.2%; PC2: 21.1%; Fig. 2). PC1 represents a variable of overall integration (based on higher clustering, lower path length) and complexity (higher density) (Fig. 2, inset), and all cetacean skulls score positive along the PC1, being clearly more integrated in comparison with other tetrapod groups (dinosaurs, sauropsids, turtles, synapsids, and squamates), but less integrated than some mammals and some amphibians. PC2 mostly sorts the skulls according to their heterogeneity vs. their complexity (Fig. 2, inset), and most derived cetaceans are placed in the morphospace with higher heterogeneity values. Thus, the skulls of most neocetes occupy a previously unoccupied region of the tetrapod skull morphospace, with better integrated and more heterogeneous skulls in comparison with other tetrapods (except some birds as Anser). Besides, cetaceans explore a quite different morphospace in comparison with other mammals, mostly because they have more heterogeneous skulls. Pakicetus and “Protocetidae” are placed close to the region of the morphospace occupied by other amphibious/semiaquatic tetrapod forms, with skulls that are less integrated (by C, especially Pakicetus) and more complex (D) in comparison with most crown cetaceans. These results reflect not only a decrease in the numbers of connections of skull bones in crown cetaceans, but also the acquisition of more irregularly distributed connections of some of these bones (increased H) within the network, suggesting a new level of organization of the cetacean skulls in their transition to the living lineages. PERMANOVA analysis supports this conclusion, showing a statistically significant difference between the skulls of aquatic and terrestrial tetrapods (p = 0.0018), between aquatic and amphibious (p = 0.0132) and between cetaceans and non-cetacean mammals (p = 0.0001). Practically, and with the exception of amphibian skulls, the skull of cetaceans represents a unique connectivity pattern (Table S2).

Skull networks specializations of cetaceans

A more detailed PCA analysis focused only on cetaceans (including an additional variable of modularity, Parcellation); it showed that the first two PCs explain nearly 70% of the recorded variation (PC1: 47.37%; PC2: 21.19%; Fig. 3). Pakicetus presents a mostly typical mammal skull network, with less integrated (especially by C) and a more homogeneous (H) skull. Protocetidae plot in an unoccupied area of cetaceans morphospace, being more complex, better integrated (by C) and slightly more heterogeneous (H) in comparison to Pakicetus (Figs. 3B–3C). This might reflect the particular skull anatomy of these stem cetaceans, unparalleled in any group of living cetaceans. In contrast, Dorudon is recovered in the morphospace of odontocetes, with a more simple (D) skull in comparison to other stem cetaceans.

Figure 3 A detailed PCA of the skull networks of cetaceans, based on the sampling herein.

(A) Sampling includes stem Cetacea, Odontoceti, and Mysticeti. The first two PCs explain nearly 70% of the variation and permit separating the skulls of cetaceans based on their heterogeneity, integration (based on Clustering and Path Length), complexity (based on Density), and modularity (based on Parcellation). All sampled mysticetes are placed in the morphospace defined generally by simpler skulls (first and fourth quadrants), further divided into those also having integrated and more modular skulls (e.g., Eubalaena and Balaenoptera spp.) or those with more heterogeneous and less integrated skulls (e.g., Piscobalaena and Aetiocetus). All stem cetaceans and most odontocetes are placed in the morphospace defined by skulls that are, comparatively, more complex (second and third quadrants), further divided into those with better integrated and more homogeneous skulls (e.g., Physeter and “Protocetidae”) and those with less integrated and less modular skulls (e.g., Waipatia and Dorudon). However, odontocetes display the greatest morphological variation. (B–D) Whereas both odontocetes and mysticetes have similar integration (albeit odontocetes display a broader spectrum), mysticetes are clearly distinguished by more heterogeneous and modular skulls, compared to the more complex skulls of odontocetes. Silhouettes have been downloaded by phylopic.org under the following credits: Pakicetus (Conty, CC-BY), Dorudon, Aetiocetus (M. Keesey, public domain), “Protocetidae” (N. Tamura, vectorized by M. Keesey, CC-BY), Physeter (M. Michaud, public domain), general Odontoceti, Tursiops, general Mysticeti, Eubalaena, Balaenoptera, Caperea (C. Huh, CC-BY-SA). Abbreviations: Ae, Aetiocetus; Al, Albertocetus; Ba_ac, Balaenoptera acutorostrata; Ba_bo, Balaenoptera borealis; Ba_br, Balaenoptera brydei; Ba_ed, Balaenoptera edeni; Ba_mus, Balaenoptera musculus; Ba_om, Balaenoptera omurai; Ba_ph, Balaenoptera physalus; Ba_ri, Balaenoptera ricei; Ca, Caperea; Do, Dorudon; Eu, Eubalaena; No, Notocetus; Pa, Pakicetus; Phy, Physeter; Pi, Piscobalaena; Pro, “Protocetidae”; Tu, Tursiops; Wa, Waipatia; Ya, Yamatocetus.

The skull of crown cetaceans explores two different and nearly completely separated morphospaces, based on H, C, L, D and P values (Table 1). Most of the extant neocetes are distinguished from extinct forms by having increased modularity (P) of the skull (except Physeter), while the other network metrics show variations within the different neocete lineages (Figs. 3B–3D). Mysticetes form a group of points that are mostly distinct from odontocetes (with a small overlap) and stem cetaceans by having better integrated (C), more simple (D), more heterogeneous (H) and more modular skulls (P) (Figs. 3B–3C). Extant baleen whales Eubalaena, Caperea and Balaenoptera spp. plot separately from the extinct toothless mysticete Yamatocetus by having comparatively more integrated (C) skulls. Yamatocetus has the most heterogeneous (H) and least integrated (C) skull, whereas the extinct toothed mysticete (Aetiocetus) has the least integrated (L) and least complex (D) skull. Within Balaenoptera different connectivity patterns are observed: B. musculus/B. ricei and B. physalus have more modular (P) skulls in comparison to other Balaenoptera species; B. musculus represents the extreme case of increased heterogeneity (H) and integration (L) within this genus, whereas the skull of B. physalus reaches the highest integration with the surrounding (C).

On the other hand, odontocetes occupy a broader morphospace, forming a group of points from negative to positive values along PC1 (between −3.5 and 1.5) reflecting a great variation of skull network organization. As a general pattern, the skull of odontocetes is more complex in comparison with mysticetes (Fig. 3B); however, both groups show a similar (and remarkable) increase in the integration of several bones with their immediate surroundings (C). The enlargement of the odontocete morphospace is expected as this group exhibits great anatomical variability in the facial skull configurations, with Physeter plotting far apart from the remaining odontocetes (on quadrant II; Fig. 3A) and the opposite position for the stem odontocete Albertocetus on quadrant IV. Practically, odontocetes demonstrate at least four different types of connectivity pattern: (i) Physeter and Notocetus have more complex (D) and homogeneous (H) skulls; (ii) Waipatia has a more heterogeneous (H) but less integrated (by C) and less modular (P) skull; (iii) Tursiops has intermediate values of heterogeneity (H), integration (C), modularity (P) and complexity (D); and finally, (iv) Albertocetus has a more modularized (P), the least integrated (by L) and the least complex (D) skull.

The PERMANOVA test only shows a statistically significant difference between the skulls of mysticetes and stem cetaceans (Fig. 3, inset).

Figure 4 Network analysis of the cetacean skull at the individual bone level, based on selected network descriptors of the individual bones.

(A–D) Violin plots with included box plots of the Degree (A), Clustering Coefficient (B), Harmonic Closeness Centrality (C), and Betweeness Centrality (D) of all skull bones of stem cetaceans (green), odontocetes (orange), and mysticetes (blue). (E–J) scatter plot of the Clustering Coefficient, Harmonic Closeness Centrality, Betweeness Centrality and Degree of the main bones involved in the two types of telescoping odontocetes (orange) and mysticetes (blue). Stem cetaceans (green) are also included for comparison.

Integration of bones within the networks

Overall, all the bones of cetacean skulls show a similar number of connections (Degree; Fig. 4A; Fig. S1); however, mysticetes and odontocetes have two or three more connections (i.e., some bones in mysticetes and odontocetes reach 14/15 connections) in comparison with stem cetaceans. Whereas the median of connections is roughly similar between the groups, the distribution of the connections is different between the two Neoceti clades. Mysticetes show more bones with low (3–4) and intermediate (5–7) number of connections, compared to odontocetes with more bones with intermediate (5–7) and high number of connections (7–14) (Fig. 4A; Fig. S1). The frontals are, in both groups, those bones with a high number of connections (12–13), followed by the vomer with more connections in odontocetes compared to mysticetes (14 over 12) (Fig. S1). The supraoccipital has the highest number of connections in mysticetes (15 in B. musculus), followed by odontocetes (11 in Physeter); however, both cases represent outlier conditions. The integration of the bones with their surroundings (CluC) shows (Fig. 4B; Fig. S2) that both mysticetes and odontocetes have bones in their skulls that are better integrated (e.g., jugal and lacrimal) or less integrated (e.g., orbitosphenoid) compared to those in stem cetaceans skulls. CloC does not show major differences between mysticetes and odontocetes, although the latter have, comparatively, more bones that are closer to each other compared to mysticetes (Fig. 4C; Fig. S3). Bones with high CloC in neocetes are the frontal, the vomer, the maxilla, and the supraoccipital (Fig. S3). The higher BetC values of the skull of both odontocetes and mysticetes (Fig. 4D; Fig. S4) indicate that neocetes have bones with more central positions compared to those in stem cetaceans.

When we compare the individual metrics of the bones that are mainly involved (both directly and indirectly) in the telescoping process (supraoccipital, frontal, parietal, vomer, maxilla, premaxilla, and nasal), some interesting observations emerge (Figs. 4E–4J). These allow tracing the different types of telescoping in odontocetes and mysticetes to the connectivity pattern of the individual bones. The clearest, and statistically significant, separation between the two groups is found in the vomer (Fig. 4G). The vomer of odontocetes has more connections (12–14) compared with the vomer of mysticetes (10–12), and it is also much more integrated with its surroundings (CluC) (Figs. S1–S2). On the other hand, the vomer is the most central bone in the odontocete Albertocetus (BetC = 172.87), although the vomer of the mysticete Piscobalaena is quite close as well (BetC =152.97). The greater integration of the vomer in odontocetes reflects the retrograde type of telescoping: as pairs of bones that are directly connected to the vomer (e.g., premaxillae and maxillae) move posteriorly and gain connections, the integration of the vomer increases. By contrast, maxillae and premaxillae do not show clear differences in the connectivity of the skull between mysticetes and odontocetes, possibly reflecting the anatomical versatility that those bones present in the different lineages analyzed.

The supraoccipital, which is the main bone involved in the prograde telescoping of mysticetes, gains importance as reflected in the higher values of CloC and BetC in extant mysticetes (Fig. 4E; Figs. S3–S4).

The frontals are bones that gain additional connections under both types of telescoping (Fig. S1). However, the acquisition of additional connections of pairs of anterior bones in retrograde telescoping slightly increases the integration (CluC) of the frontals in odontocetes, while they occupy a similar central position in both odontocetes and mysticetes (Fig. 4F, Fig. S4). Another interesting observation is the placement of the left and right frontals of Physeter in two different quadrants (compared to the symmetric skulls, where both bones are at the same point), meaning that the strong asymmetry of the skull observed in Physeter creates different connectivity patterns for the same bone on each side. A similar result is seen in the premaxilla (Fig. 4I), but not in the maxilla (Fig. 4H), where right and left bones are placed in the same quadrant.

Organizational modularity of skull cetacean networks

The detection of modules in anatomical networks is a matter of ongoing debate (see Esteve-Altava, 2020) and references therein). In general, the skulls of neocetes are more modular (P) compared with stem cetaceans (Table 1). Extant mysticete skulls are more modularized in comparison with extant odontocetes, with increased modularity in Balaenoptera spp. (P = 0.738–0.797) and a remarkable decrease in Physeter (P = 0.666). The best modularity solutions consistently recover four main modules: two dorsolateral, one palatal and another one in the posterodorsal region in stem cetaceans (e.g., Dorudon), mysticetes (e.g., Yamatocetus), and odontocetes (e.g., Notocetus), in both symmetric and asymmetric reconstructions (Fig. 5). In some cases, the posterodorsal module could be divided into left and right portions (e.g., Eubalaena and Caperea; Fig. 5). Given the various issues in the reconstruction of the modules (see Esteve-Altava, 2020), we refrain from discussing their boundaries in detail.

Figure 5 Anatomical networks, recovered network modules and the evolution of the main network descriptors under parsimony in a phylogenetic framework.

The phylogeny is based on Martínez-Cáceres, Lambert & De Muizon (2017) for stem cetaceans, Marx et al. (2019) for mysticetes and Viglino et al. (2021) and Boessenecker, Ahmed & Geisler (2017) for odontocetes. See supplementary information for detailed mapping.

Discussion

Telescoping promotes a new path in the connectivity of living cetacean skulls

During their transition to the aquatic environment, cetaceans experienced profound changes in their skeletal anatomy, especially in the skull. Among the most remarkable are the posterodorsal migration of the external bony nares, the reorganization of the skull bones (= telescoping), and the extreme cranial asymmetry (characteristic of odontocetes) (Miller, 1923; Fordyce & De Muizon, 2001; Berta, Ekdale & Cranford, 2014; Marx, Lambert & Uhen, 2016). Cranial telescoping represents an important key innovation in the evolution of Neoceti and might be linked to facilitating breathing while they are submerged, structural reinforcement of the vertex to avoiding fractures during the air-breathing movements, and the acquisition of filter-feeding in mysticetes and echolocation in odontocetes (e.g., Miller, 1923; Fleischer, 1976; Heyning & Mead, 1990; Oelschläger, 1990; Churchill et al., 2018; Roston & Roth, 2019). It represents an important anatomical shift in the topological organization and sutural contact of cranial bones (Miller, 1923; Roston & Roth, 2019), and thus in the connectivity of the skull elements, breaking the mold of the mammalian skull. Our study is the first attempt to analyze the patterns of skull connectivity in cetaceans captured through the lens of anatomical networks.

Our results show that, along with the transition to a fully aquatic lifestyle, the cetacean skull underwent a remarkable reorganization of the connectivity pattern that allowed the exploration of a new tetrapod morphospace. While stem cetaceans (especially Pakicetus and “Protocetidae”) still remain in the known morphospace for other non-cetaceans mammals, with comparatively less integrated and more complex skulls, most living cetaceans studied here follow the path towards more heterogeneous (H), better integrated (C), and simpler skulls (Fig. 5; Figs. S5–S11). Despite telescoping promotes contacts between bones that otherwise would not be possible (e.g., occipital and rostral bones), the number of connections and thus the complexity (D) of the skull networks decreases at the base of Neoceti. The bones that reach the widest range of variations in the number of connections are the supraoccipital (5–15), frontals (6–13), maxillae (5–11), but also the pterygoids (4–9), presphenoid (3–9), basisphenoid (3–9), alisphenoid (2–6), and ethmoid (2–9) (Data S2; Fig. S1). These results suggest that the rearrangement of facial and occipital bones impacts not only the number of connections of those bones directly involved in telescoping (e.g., maxilla and supraoccipital), but also of other bones (e.g., palatal bones) not directly affected by telescoping (see further discussion below).

One of the network descriptors that better define the evolution of the cetacean skull is heterogeneity, which shows an important increase at the base of the Pelagiceti (a clade including basilosaurids and kekenodontids plus living cetaceans), and even further in the Mysticeti clade (Fig. 5 and Fig. S6). In terms of anatomical networks, heterogeneity reflects a disparity in the number of connections among the skull bones, indicating different hierarchy levels of the parts of a network (i.e., anisomerism; Esteve-Altava et al., 2013a; Rasskin-Gutman & Esteve-Altava, 2014). In tetrapods, the increase of the specialization of individual bones has been linked to the appearance of new unpaired bones by fusion of paired ones (Esteve-Altava et al., 2013a). Cetaceans do not present variations in the number of bones by loss or fusion in the different groups analyzed (except the condition of interparietal in neocetes and Physeter which lost one nasal; Fig. S7). However, our results show that the unpaired bones ratio (a measure of anisomerism; Esteve-Altava et al., 2013a) is higher in cetaceans in comparison with other tetrapods that have the same number of bones (Fig. S12), suggesting an increase in the specialization of individual bones. We hypothesize that telescoping provides an additional mechanism (as bone loss or fusion) to increase the connectivity pattern of unpaired bones and thus increase the heterogeneity of the skull networks.

Another hallmark path that marks the evolution of the connectivity pattern of neocetes is the increase of the integration of the skull (C) (Fig. 5 and Fig. S10). Our results show that clustering appears to be a good descriptor of telescoped skulls, reflecting an increasing integration of the bone elements of the skull in this new level of organization. The bone overlap and the proximity of occipital-rostral elements affect directly the connectivity of those bones involved in the telescoping process, but indirectly also affect the relationship of other bones. For example, at the bone level there is an evident increase in the integration (by CluC) only in the premaxilla and parietal; however, palatal bones, such as the palatine, pterygoid and vomer, but also the presphenoid and alisphenoid increase their integration in comparison to stem cetaceans (Fig. S2). The topological reorganization of palatal bones during telescoping, mainly the covering of the palatine and alisphenoid by the pterygoid, has been suggested in the pioneering work of Miller (1923) but not extensively studied in modern analyses (e.g., Churchill et al., 2018; Roston & Roth, 2019). Besides, changes in the distribution and contacts of these bones along with the evolution of different cetacean clades, but not directly linked with the telescoping process, have also been reported (Muller, 1954; Fraser & Purves, 1960; Bouetel & De Muizon, 2006). In addition to the telescoping process, the skull of neocetes has experienced profound changes associated with the development of air sinus and vascular systems and the modifications in the ear region (even more evident in odontocetes due to the evolution of the echolocation system) (Fraser & Purves, 1960; Reidenberg & Laitman, 2008; Mead & Fordyce, 2009). In particular, the air sinus and vascular system develops in the basicranium and orbital region, extending mainly over the surface of pterygoids, palatines, basisphenoid, and alisphenoids, with variations in their development and in the configuration of their bone-correlates among the different groups of neocetes (see Fraser & Purves, 1960 for a detailed analysis). Besides, changes in the orientation of some basicranial elements (i.e., presphenoid) associated with repositioning of the nasal passages (i.e., basicranial retroflexion) occurring during the prenatal ontogeny also contribute to the rearrangement of skull elements (Roston & Roth, 2021). We speculate that the increase in the integration of the neocete skull was achieved not only in relation to the changes associated with the telescoping process but also with all the morphological modifications that occur in the basicranium and palate linked with the specialization to underwater hearing, echolocation, breathing, and deep diving (Fraser & Purves, 1960; Cranford, Amundin & Norris, 1996; Cranford et al., 2008).

In addition to the increase in the integration (C) and heterogeneity in neocetes, there is a marked shift toward an increment of modularity (P) from non-telescoped to telescoped skulls (Fig. 5; Fig. S11). Connectivity modules differ from variational modules—sensu Esteve-Altava, 2017—in that they reflect the topological arrangement of anatomical units, not their shapes; thus, information of connectivity modules should be presented as a complement of the information generated with the variational modules (Rasskin-Gutman & Esteve-Altava, 2014; Esteve-Altava, 2017). Unfortunately, studies on the variational modules in cetacean skulls are very scarce and only focus on odontocetes (Del Castillo et al., 2017; Churchill et al., 2018). The number of modules identified depends if the models of modularity identified are based on a development correlation (in which case 3 modules were identified; Del Castillo et al., 2017) or on a functional correlation (between 5–10 modules; Churchill et al., 2018). Our analysis shows a mean of four connectivity modules for neocetes, with a variable number in odontocetes (between 3–5) and a more constant number in mysticetes (between 4–5 modules), associated with the rostral and orbital (recovered in symmetric modules), basicranial (including in a variable array of the bones of the floor of the cranium as well as palatal bones), and cranial regions (including the bones that form the cranium and the squamosals). These organizational modules are more closely related to the basicranium, neurocranium, and rostrum modules reported by Del Castillo et al. (2017), and suggest a basic connectivity modularity pattern of the neocete skull. Due to telescoping that promotes new sutures (e.g., premaxilla and frontal, frontal and supraoccipital) and new types of sutures (= horizontal sutures; Gatesy et al., 2013; Roston & Roth, 2019), it is likely that telescoped sutures more than bones themselves, mark an important constraint in the topological arrangement of the anatomical units and, thus, in their connectivity. Underlying developmental processes (of both bony and soft structures), structural constraints on shape, bone growth, and/or biomechanical functions are probably the main parameters responsible for the origin of connectivity modules (Esteve-Altava, 2017).

The impact of telescoping on mysticete and odontocete skull networks

Two main patterns of telescoping can be traced in mysticetes and odontocetes, with important differences in the topographical organization of the skull bones. In mysticetes, telescoping is dominated by the forward movement of the supraoccipital and parietal until the orbit level, while only a narrow medial part of maxilla extends posteriorly, interlocking with the frontal (but not covering it). In odontocetes, rostral bones (maxilla and premaxilla) extend backwards, approaching the supraoccipital; in this case, the maxilla spreads over almost the whole surface of the frontal, including the supraorbital process (Miller, 1923; Fig. 1). Different developmental sequences of bone ossification and suture closure have been identified as the underlying processes that influence the skull anatomy of both groups (e.g., Perrin, 1975; Lanzetti, 2019). What is the impact of these disparate skull anatomical organizations of odontocetes and mysticetes in the network organization?

Prograde telescoped skulls of mysticetes differ in their increased heterogeneity (H), modularity (P) and integration (C), while retrograde telescoped skulls of odontocetes follow the path of increasing complexity (D), reaching the highest number of total connections (Fig. 5; Figs. S5–S11). The integration by proximity (L) shows an increase at the point of diversification of crown mysticetes, whereas in crown odontocetes this integration remains without changes. This suggests that one of the main characteristics of the telescoping process—shortening of the occipital-maxilla distance—impacts the integration by proximity of the skull network in a different way, depending on the telescoping specialization followed by each group. However, this hypothesis should be further tested with an expanded sample of odontocetes.

In mysticetes, the increased anisomerism (H) is the main connectivity pattern of the prograde telescoped skull. An irregularity in the numbers of connections of the bones is evident in mysticetes (see the variations in the numbers of connections of e.g., alisphenoid, presphenoid, supraoccipital, and frontal; Fig. S1). As expected, the supraoccipital gains relevance and centrality (BetC) in the networks of extant mysticetes and also achieves a high number of connections (e.g., 15 connections in Balaenoptera musculus) if we compare it with an archetypal odontocete skull as the one of Tursiops (Physeter reaches 11 connections; however, the morphology of its skull is quite disparate from other odontocetes due to the extreme posterior extension of the maxillae, the evident asymmetry of the premaxillae, and the lack of one nasal; see further discussion below) (Data S2). On the other hand, the increased connections (93–97) and, thus, complexity (D) in the skull network of crown odontocetes are probably linked to the dominant movement of paired bones in retrograde telescoping, such as the maxilla, which gains connections (Dec) and integration not only in its own node but also in its surroundings (CluC). No remarkable differences are observed between odontocetes and mysticetes (especially in extant forms) in the connections/integration/centrality of other bones also affected by telescoping, such as the premaxillae, frontals, nasals (except CluC), and parietals. On the contrary, the vomer, alisphenoid, presphenoid, and pterygoids show conspicuous differences between both groups in the numbers of connections, centrality (BetC), and integration. This result provides evidence that, again, even though telescoping defines the quite distinct anatomical configuration of the skull of mysticetes and odontocetes, there is not a broad effect in the connectivity pattern of all the bones directly involved in these processes. Our results invite us to re-evaluate the role of palatal and sphenoid bones in the evolution of the skull of neocetes and we may consider them as the “hidden hands” that play a key role in the improvement of connection and integration of the different elements of the skull. Future works should focus on analyzing with more detail the anatomical reorganization of these regions, and their correlation (or lack of it) either with the telescoping, with the basicranium retroflexion (Roston & Roth, 2021), with the evolution of air sinus and vascular systems, or with a combination of all of these processes.

Skull connectivity patterns within mysticetes

Within mysticetes, some particularities can be identified in the skull networks of the different lineages (Figs. S5–S11). The toothed mysticete Aetiocetus presents the smallest number of connections of the whole mysticete sample and the simpler (D) and least integrated (by L) skull. This pattern of skull connectivity is consistent with the poorly telescoped skull of this taxon, reflected in a non-telescoped supraoccipital, and a broadly exposed parietal and frontal in the skull roof (e.g., Deméré & Berta, 2008). Furthermore, the skull of the toothless mysticete Yamatocetus differs from Aetiocetus in the more pronounced telescoping of the supraoccipital, but still retains a long intertemporal region. These small anatomical changes might explain the slight increase in the complexity (D), integration (by L), and heterogeneity (H) observed in Yamatocetus in comparison with Aetiocetus.

Within the crown mysticetes prograde telescoping reaches a more advanced state in comparison to stem forms and noteworthy anatomical variations can be observed among the different lineages. Balaenids and neobalaenines have a telescoping process dominated by the pronounced anterior expansion of the supraoccipital, which extends beyond the level of the orbit and excludes the parietal from the vertex of the skull. Besides, the nasal and ascending process of the premaxilla and maxilla do not protrude into the occipital region, defining a sub-rectilinear suture between frontal and maxilla (Miller, 1923 pl: 8; Bouetel, 2005). Instead, in balaenopterids both rostral and occipital elements move in similar proportions: maxilla, premaxilla, and nasals project backwards, until the half-length level of the orbit, while the supraoccipital extends forward, meeting the rostral bones at almost the same level (Miller, 1923; pl 8,5). This configuration is responsible for determining a strong interdigitation of frontals and maxillae, which has an important biomechanical function supporting the forces induced during lunge feeding (Lambertsen, Ulrich & Straley, 1995; Bouetel, 2005). Extinct cetotheriids show a pattern of telescoping similar to balaenopterids, with a more extreme posterior extension of the maxilla (beyond the level of the orbit) that limits the supraoccipital to the posteriormost portion of the skull. Crown mysticetes achieve the highest skull integration (by C and L) of all mysticetes (Fig. 5), suggesting that, regardless of the anatomical variations observed in the prograde telescoping, the improvement of the anatomical integration is a distinct hallmark of this group of neocetes.

Among extant lineages, the telescoping process of balaenids and neobalaenines (Caperea) appears to be related to an increased structural disparity (H) and modularity (P) of the skull networks. In Caperea the supraoccipital has a more extreme forward expansion, assuming a more central role in the skull network; however, the number of connections does not change in comparison to balaenids. The increased heterogeneity observed in the skull networks of these mysticetes are reflecting a disparity of roles for the different parts of the network, which might relate to different morpho-functional correlations. In balaenids and neobalaenines the unusual skull architecture is also strongly influenced by the highly specialized skim feeding method of these whales (Werth, 2004; Bouetel, 2005). It is possible that the evolution of heterogeneously connected skulls in these lineages has been driven not only by the distinct telescoping process but also by structural constraints imposed by their feeding behaviour.

Within an evolutionary context, the skulls of Balaenoptera spp. are the most homogeneous (H), less complex, and best-integrated with the surroundings (C) in comparison with other extant mysticetes analyzed (Figs. S6–S11). Balaenopterid telescoping promotes skull networks with similar structural connections, since there is no specialization of some elements over others, but reaching a better integration by interdependence (C) among the nodes of the network. Within Balaenoptera, conspicuous variations in the shape and contacts of some bones are observed in the vertex, which has been used as a source of diagnostic characters to define species (see for example Wada, Oishi & Yamada, 2003; Yamada et al., 2006). Our results show that this interspecific variation impacts the connectivity pattern of the skull, but in a lesser extend compared to all the variation seen in Mysticeti. For example, the skull of B. musculus has an extreme posterior elongation of the rostral bones and frontals are not exposed behind the nasals, resulting in an advanced condition of telescoping in comparison to other Balaenoptera species (and even to other mysticetes). This condition is probably related to the “shorter skull” (i.e., the best-integrated by L) observed in this species. Besides, the supraoccipital gains connections from paired rostral bones (Fig. 4), thus increasing the complexity (D) of the skull network (Table 1). In the remaining species, where different vertex configurations are observed (especially related to the exposition of the frontals), there are small variations (except in P) in the network descriptors, which might reinforce the idea that not all the anatomical variations impact the skull connectivity pattern.

Skull connectivity patterns within odontocetes

Within odontocetes, the skull connectivity pattern differs between stem and crown lineages (Fig. 5; Figs. S5–S11). The skull network of Albertocetus, a stem odontocete (e.g., Uhen, 2008; Churchill et al., 2016), shows the smallest number of connections (K) for the whole odontocete sample (Table 1), together with the least complex and least integrated by proximity (L) skull. This is probably related to the relatively less advanced stage of telescoping observed in Albertocetus, as evidenced by the short posterior projection of the ascending process of the premaxillae, poor lateral expansion of maxillae, and broad exposure of the parietals and frontals in the roof of the skull (phase one sensu Churchill et al., 2018). Crown odontocetes are distinct by having an increasing number of connections (K) within the skull networks and, thus, more complex skulls (D) (Fig. 5; Figs. S5, S8). Platanistoids (sensu Viglino et al., 2021) represented by the extinct forms Waipatia and Notocetus, have less modular skulls in comparison to Tursiops. In addition, these taxa exhibit a mosaic in their network skull descriptors, with density (D) and integration by proximity (L) being close to the values of extant odontocetes, while integration by clustering (C) and heterogeneity (H) represent extreme and unique values—i.e., Waipatia is characterized by the least integration of bones with their surroundings (C) among all odontocetes analyzed here while Notocetus, on the opposite side, represents the most homogeneous odontocete (H). A more advanced stage of telescoping is present in platanistoids, with an almost absent intertemporal region and a more pronounced posterior expansion of maxillae, premaxillae, and nasals (phase II of Churchill et al., 2018). This progress in telescoping is reflected in the increase of the number of connections (K), complexity (D), and integration (L) of the skull in comparison to the stem odontocete Albertocetus. Nevertheless, our results suggest that there is not a clear skull connectivity pattern for platanistoids, reflecting the variety of skull morphologies and feeding strategies observed in this group (Viglino et al., 2021). Future AnNA, including an expanded sample of platanistoids, could further test this hypothesis.

Finally, the two analyzed extant odontocetes (in the families Delphinidae and Physeteridae) show disparate patterns of skull connectivity, especially Physeter with its bizarre cranial morphology (Figs. S5–S11). While Tursiops represents a more archetypical stage of retrograde telescoping (i.e., with a broad overlap of the maxilla and frontal bones; Fig. 1), Physeter has an extremely telescoped skull, with a highly asymmetrical facial region, and the loss of one skull bone (Flower, 1868, figs.1-2). While the skull network of Tursiops is distinct with its highest anisomerism (H) and modularity (P), Physeter has the most complex (D), the best integrated (by L and C), and the least modular skull of all the crown odontocetes sampled. The better integration by proximity (L) of the skull of Physeter is not unexpected due to the pronounced shortening of the occipital-maxilla distance and, thus, the gain of bone contacts—in for example the maxillae and supraoccipital (Fig. 4)—, as well as for the increase of complexity related to the loss of one nasal bone (as suggested by Esteve-Altava et al., 2013a for the evolution of tetrapod skulls). Considering that the skull morphology of Physeter is constrained not only by its particular telescoping process but also by the distinct soft tissue structures related to its highly specialized echolocation system (Huggenberger, André & Oelschlaeger, 2016), the connectivity pattern of this skull may be the result of combined underlying processes.

A better integration of the bones with their surroundings (C), probably enhanced by the backward movement of paired bones, as well as a more modular skull (P) are preliminary suggested as a distinct connectivity pattern of the retrograde skull of extant odontocetes. Future works, including a broader sample of extant odontocetes, could further test this hypothesis.

Conclusions

Telescoping is one of the most remarkable changes in the anatomy of the cetacean skull and it has been associated with a plethora of morpho-functional explanations. Along with the topographical re-organization of the skull bones, our study shows that telescoping also promotes profound changes in the connectivity pattern and integration. Living cetaceans (Neoceti) explore a new morphospace in comparison to other tetrapods (and even to other mammals), with better integrated, slightly simpler, and mainly more heterogeneous skulls. This represents a break in the mammalian skull mold, triggering the exploration of new ecological and behavioural strategies. Our study provides further evidence that not only the bones directly involved in the telescoping process (e.g., supraoccipital and maxillae) gain relevance and integration in the skull networks of neocetes, but also other bones (i.e., vomer and presphenoid) that are not obviously affected by telescoping. It is possible that telescoping together with all the basicranium specializations linked with hearing/breathing/deep diving adaptations are mainly responsible for the changes in the connectivity pattern of neocete skulls.

Distinct skull connectivity patterns were identified in mysticetes and odontocetes, with prograde telescoped skulls of mysticetes being characterized by an increased heterogeneity and modularity, while the retrograde telescoped skulls of odontocetes are characterized by greater complexity. Besides, retrograde telescoping causes increased integration in the maxillae of most odontocetes, while prograde telescoping of mysticetes promotes greater importance and centrality of unpaired bones (i.e., the supraoccipital). In odontocetes, the asymmetry of the skull triggers different connectivity patterns for the same bone on each side.

Particular connectivity patterns of the skull were preliminarily identified within the different lineages of odontocetes and mysticetes analyzed here. We found that major anatomical changes impact the connectivity pattern of the skulls (i.e., those associated with different styles of telescoping), whereas others (i.e., interspecific variation in the skull vertex of Balaenoptera) remain almost invisible through the lens of the AnNA. Along with feeding, hearing, and echolocation specializations, telescoping could have driven the evolution of the different connectivity patterns of living cetacean lineages.

Finally, our results show that not all shape variations observed along the evolution of the cetacean skull have a direct impact on the topological organization and connectivity of the elements of this complex structure; this reinforces the idea that Anatomical Network analyses are a complementary tool to the other areas of morphological research, which need to be further explored (e.g., with an expanded sample and/or adding information on soft tissue anatomy).

Supplemental Information

Supplemental Information 1 Network data

Click here for additional data file.

Supplemental Information 2 Metric of the individual bones, matrix and PCA data

Click here for additional data file.

Supplemental Information 3 Supplemental Tables

Click here for additional data file.

Supplemental Information 4 Supplemental Figures

Click here for additional data file.

We thank Florencia Paolucci (CONICET-MLP) and Mariana Vilgino (CONICET-CENPAT) for helpful discussions in the construction of anatomical networks of Physeter and Notocetus, and Marta Fernández for their suggestions in the draft version of the manuscript. We would like to thank Enrique Crespo and Néstor Garcia (CESIMAR-CENPAT) for access to extant odontocete specimens. We also thank Anahi Formoso (CENPAT) for the revision of the English grammar. We thank the reviewers (O. Lambert, M. Churchill and an anonymous reviewer) and the editor (N. Pyenson) for their thoughtful and useful comments on this manuscript.

Additional Information and Declarations

Competing Interests

Author Contributions

Data Availability

The authors declare there are no competing interests.

Mónica R. Buono conceived and designed the experiments, performed the experiments, analyzed the data, authored or reviewed drafts of the paper, and approved the final draft.

Evangelos Vlachos conceived and designed the experiments, performed the experiments, analyzed the data, prepared figures and/or tables, authored or reviewed drafts of the paper, and approved the final draft.

The following information was supplied regarding data availability:

The raw data are available in the Supplemental Files.

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
