# Peer review of "Breaking the mold: telescoping drives the evolution of more integrated and heterogeneous skulls in cetaceans"

_PeerJ, doi:10.7717/peerj.13392_

## Round 0.1 · original submission · Major Revisions

This manuscript has now received three excellent reviews, all of which recommend revisions to varying degrees. The preponderance of feedback points to more towards major revisions than minor ones (i.e., moderate revisions), although in my view the identified revisions are not so substantive that they require full-scale re-analysis or rethinking of the manuscript.

One the minor side, there are grammatical and prose concerns that require simple edits for clarity and readability; towards the more major side, the authors should heed the reviewers calls (especially those of Reviewers 2 and 3) for stronger delineation of the ideas that underpin this paper: telescoping, modularity, and integration are all separate concepts that should be clearly stated and bounded. Also, the authors should focus on replying to all of Reviewer 3's detailed comments, including those that ask good questions of the analyses and datasets.

I think all of these concerns can be addressed, and they will make the paper better. Overall, this is an interesting and important contribution, using new methods, to a century-old problem in cetacean biology that has escaped full explanation. I look forward to seeing the revisions, which will be sent back out to review.

·

Excellent Review

This review has been rated excellent by staff (in the top 15% of reviews)
EDITOR COMMENT
Fantastic, detailed, and balanced feedback from a specialist. This review goes a long way to showing how deep expertise can be receptive towards new advances that have direct application for understanding broad evolutionary questions in a topic.

Basic reporting

- The text is written in clear and professional English. I made some suggestions in the annotated pdf for minor changes (typos, style) that may slightly improve the reading. However, I am not a native English speaker, so authors should feel free to choose for alternate formulations.

- The scientific background is adequately presented, with a sufficient number of references. The authors clearly demonstrated an excellent knowledge of the general context and research questions.

- The text is properly organized. It is illustrated by very fine figures, all clear and useful. Figure captions provide enough details (I only made a few suggestions for the addition of some information. I am not familiar with the analyses performed here, but to my knowledge all the raw data has been made available with the ms at submission.

Experimental design

- Though originating from previous work on other tetrapod groups, this work focuses on a newly acquired dataset targeting a series of extinct and extant cetaceans, allowing for the discussion of new research questions with a cutting-edge technique.

- The research questions are clearly presented, they focus on topics that have been barely tackled in a quantitative way in the past, and results are adequately articulated with the introductory questions.

- As mentioned above, I am not familiar with the analytical tools implemented here. Methods are generally explained with sufficient detail, though in a few cases I suggested in the annotated pdf the addition of a few words (on data acquisition and some of the parameters of the recovered networks). This may help other researchers interested in this technique to apply it and to better apprehend the interpretations.

Validity of the findings

- The data supporting the interesting conclusions presented here are made available. Though one may expect an even broader taxonomic sampling for future work on this topic, the chosen taxa have been carefully selected and allow for the discussion of interesting evolutionary steps. It may be useful to add some details on why extinct taxa were selected, even if not all extant cetacean families have been sampled. I find this sample very fine for such a first analysis, but a few words could be added, especially considering that fossil skulls must be more difficult to investigate in some regions (palate, dorsal part of neurocranium) due to the lower visibility for suture lines.

- Most conclusions are adequately supported by the presented results. In some cases (see annotated pdf and comments below), the sample size may be slightly too small to allow for the unambiguous detection of evolutionary trends. I made a few comments about odontocetes, but it may be worth checking for other clades. In most cases my impression is that it is mostly a question of formulation, with more speculative interpretations to be more clearly identified.

Additional comments

- This is a great study, and I am glad to see this promising technique being applied to cetaceans. This really sounds like the perfect match considering trends previously detected among cetaceans. Obviously, the sample could be enlarged in future work, for example including at least one member of each modern family, but this dataset already provides the opportunity to discuss about a series of trends and adaptations. Most of my comments can be found in the annotated pdf, but a few others are listed below.

- I guess that the ontogenetic stage will have a rather strong impact on the results of such an analysis, with young individuals' rostral bones not reaching as far posteriorly in the facial region (at least for odontocetes). Could you add a few words about the ontogenetic stage of the selected specimens?

- I wonder how you dealt with ankylosed/fused bones. In some parts of the skull of aged individuals, it can be tricky to distinguish which bone(s) is(are) involved, for example in the dorsal part of the neurocranium, with contributions of the parietals/interparietal that are often difficult to assess. Could you add some words on this question? Also, it may be informative to add some words in the material and methods section about missing data in extinct species (I guess that you cannot reach the same resolution with a fossil skull compared to a modern one with all sutures visibles.

- Related to my previous comment, the position of Notocetus in the PCA (fig. 2) is very surprising. It seems way too far from Waipatia or Tursiops for example. Shouldn't the network data be checked for that taxon?

- For the more detailed PCA of cetaceans, considering the low number of odontocete taxa and the wide spacing of the different taxa in the graph I am not sure that this sample is large enough to draw firm conclusions. Indeed, you may probably expect additional taxa to further increase the surface of the morphospace occupied by odontocetes. This is just a comment, but in some places it may be worth slightly reformulating the interpretations (for example for the near absence of overlap between odontocetes and mysticetes, and for the description of several trends).

I am looking forwards to seeing this very interesting work published.
O Lambert

·

Basic reporting

Buono and Vlachos present a novel and interesting paper exploring the evolution of cranial telescoping using network analysis of cranial bones. They find distinct patterns consistent with the two modes of telescoping which dominate the two major groups of living whale, and that neocete whales in general occupy a distinct region of the tetrapod morphospace, highlighting the uniqueness of telescoping in whales.

Overall I found the paper (mostly) easy to follow with proper citations and a sensible structure, with conclusions that were well supported and explained. There are a few areas however that require improvement. Some of these are concerns over explanation of methods, which I will discuss in experimental design.

Relevant to Basic reporting, I found a fair bit of improper or odd phrasing and sentence structure, as well as incorrect use of words. These are mostly minor grammatical concerns, and when possible I do highlight cases of improvement. I would strongly recommend however additional passes through the manuscript to fix these.

Experimental design

The experimental design is appropriate and the research fits well within the aims and scopes of the journal, representing important advancements in our understanding of whale cranial evolution. I generally understood the paper, however I feel there are two significant aspects of the methods that need addressing.

First, I think more detail is needed in the methods. Anatomical Network Analysis is a fairly new method that is likely not going to be very familiar to many individuals interested in marine mammal evolution, and I feel the authors don't adequately explain the method in a way understandable for those who don't have a background in the technique. While I thought the different metrics were adequately explained, I wasn't sure how they were generated without reading up in more detail on the method. There should be a short section describing the method in more general terms, and how cranial configurations of bones are used to generate a matrix in which the listed metrics are generated.

On a related note, there needs to be an implicit statement that the different metrics are what is being used to create the PCA and PERMANOVA. My initial thoughts were that the authors were using the matrix produced in some manner to create the PCA.

I would also like to see the authors address cranial asymmetry. As the authors themselves mention, cranial asymmetry in odontocetes can significantly influence skull shape, especially in extremely asymmetrical taxa such as Physeter. Did the authors treat right and left bone separately, or were both bones considered the same bone as far as the connections go. Were their differences in metrics for some odontocetes if the right and left were treated separately? If they were not treated differently, than at least a statement of WHY is needed: Is the asymmetry just not significant enough to affect the given cranial configurations? I don't necessarily expect a giant section on this topic...just a few sentences may be all that is required to address this concern.

Validity of the findings

The results make sense and are well supported. Most of the underlying data is provided. The one thing I didn't find was the raw matrix that was used to generate the different metrics, just a table of the metrics produced. This would be useful for supplemental information, as without that raw matrix other researchers have to start from scratch to validate the results.

Now I should mention here that one of the supplemental files didn't download properly for me, and I am not sure if that is due to my computer or PeerJ. If this information was provided, feel free to ignore this comment, although perhaps double-check that the data can be exported in a meaningful manner.

Reviewer 3 ·

Basic reporting

There are a few places where the phrasing or word choice was ambiguous, and there were also many typos. I’ve tried to point these places out as I went through the manuscript.

Overall, the structure and figures meet professional standards. I have made some minor suggestions to improve clarity and organization.

The Results and Discussion are self-contained and new insights about the evolution and anatomy of cetacean skulls. But, the introduction and definition of network parameters and how they relate to network descriptors and morphology is not self-contained. AnNA is still a relatively new method with which many morphologists may not be very familiar. While I myself have read numerous papers using AnNA, I still had to look up the mathematical definitions and interpretations presented in several papers by Esteve-Altava et al. I think the present paper would strongly benefit from clearer explanations of the network parameters and descriptors, and how they relate to morphology. The authors did this later in the Discussion (e.g. line 316, heterogeneity) but the reader needs to understand these metrics much earlier in order to understand the Results.

Experimental design

This work is original and provides important new insights about how cetacean skulls compare with other mammals and tetrapods, and falls within the aims and scope of the journal.

I think the research questions can be more clearly defined. The last paragraph of the Intro was particularly confusing and hard to follow. Perhaps the authors can add explicit details about how they hypothesize/predict telescoping will affect the network and then assess whether the data support these in the Discussion.

In the methods, I would like to see how the authors dealt with interspecific and intraspecific variation. For instance, balaenopterids have considerable morphological variation at the skull vertex (Tsai et al. 2014; Nakamura et al. 2016). How did the authors account for such variation, or does that morphological variation not affect the network? Additionally, were the authors only including adult specimens, and if so, how did they know they were adults? This is an important detail, because telescoping especially could change with age (which is still unclear).

Validity of the findings

The findings are valid, but I was curious why the authors did not include more odontocetes, especially some major groups, such as ziphiids and phocoenids. Odontocetes are more speciose than mysticetes, though their sampling does not reflect that. And, these additional odontocete taxa are morphologically distinct from the ones included in the study. While I think the results are meaningful even without including a broader sampling of odontocetes, I think these choices about species sampling should be more clearly explained in the Methods.

All data have been provided, with the exception of a mislabeled CSV file for Aetiocetus.

The conclusions are well-stated, but the original research questions could have been clearer. I think it would help if the authors clearly outlined hypotheses and predictions about how telescoping would affect the anatomical network, and then explain in the Discussion if their results support or reject these hypotheses/predictions.

Additional comments

1. Clearly define odontocetes as toothed whales or echolocating whales & mysticetes as baleen whales in the Intro. This will both make the article clearer to readers unfamiliar with cetaceans and can also help clarify toothed mysticetes vs odontocetes later in the article.
2. The terminology around network descriptors and parameters seems to vary somewhat throughout the manuscript. These should be clearly defined in the Introduction and used consistently throughout the manuscript.
3. I had a few questions in the methods section. How did the authors account for interspecific and intraspecific variation within a single genus? This question especially applies to the balaenopterids, where it has been previously shown that they can have a lot of variation in the morphology at the vertex of the skull (e.g., Tsai et al. 2014, https://doi.org/10.2517/2014PR009 ; Nakamura et al. 2016, https://doi.org/10.32211/lamer.54.1-2_1). Also, were all of the specimens adults, and how was this determined?
4. Additional sub-section headers in the Discussion to break up the “Mysticete and odontocete skull network specializations” are needed. This section covers a lot of information and it would be helpful if the authors could more clearly point out when they are comparing mysticetes vs odontocetes or making comparisons among lineages within each group.
5. In figure 2, it was surprising to see some of the cetaceans in the same region of morphospace as birds. Do the authors have ideas about why this might be the case? Also, I’m curious if the authors can provide more insights in the Discussion about how cetacean skulls compare with other aquatic tetrapods.
6. Novel bone contacts (links) may not mean that there are novel sutures between those bones (i.e., no sutural ligament). In other words, the bones may be simply near each other, but not have a sutural joint between them. Can the authors provide insights about whether novel bone contacts have novel sutures, or whether the bones are only adjacent/neighboring? This question is especially relevant to lines 379-384, but also throughout the ms.

Specific Clarifications:
Abstract: When discussing integration and modularity in shape (Olsen & Miller, 1958), modularity and integration are often discussed as opposites. So, it’s confusing that cetacean skulls can be both more modular and more integrated. Could the authors clarify here how a skull can be both more modular and more integrated? Or, maybe point out in the Abstract that these terms have different meanings in AnNA, and then discuss this conceptually in the Intro?
106-107 This would be a helpful place to explain the network parameters/descriptors and how they relate to the concepts of integration, modularity, heterogeneity, and complexity.
107-110 This sentence and the following paragraph is confusing. It may be clearer to separate the results from Esteve-Altava et al. (2013a) and discuss them in the previous paragraph, and then to focus this paragraph only on explaining the current study. I also recommend that the authors outline clear hypotheses and predictions here or in a paragraph explaining the network parameters/descriptors. Given what they know about cetacean skull morphology, how do they predict it to affect the network?
137-141 This is another place where mathematical and morphological explanations of the network descriptors would be helpful. The authors did a good job providing this information for the centrality measures in lines 144-148, and I suggest they do something similar here.
140 Should all words be capitalized in Average Path Length? Also, is this the same as Mean Shortest Path Length, which is the term used by Esteve-Altava et al.? Or, is it a different metric?
201-202 In standard mathematical numbering, wouldn’t Physeter be in Quadrant II and Albertocetus is in Quadrant IV? Or, is there a different system being used here? The quadrants should be labeled in the figure to make this clearer.
271-273 It would be really helpful to have anatomical regions in some form described here, though I realize the modules don’t directly correlate with anatomical regions. For instance, is the posterolateral module in the general area around the ear or more in the area of the temporal fossa?
295 odontocetes are described elsewhere as having more complex skulls. So, this sentence seems to contradict other parts of the manuscript, saying cetaceans have evolved simpler skulls
316 It would be helpful to have this explanation of heterogeneity earlier in the manuscript (perhaps even the introduction).
324-326 “We hypothesize that telescoping...” This sentence is confusing to me. Do the authors mean that telescoping, in addition to bone loss of fusion, provides a mechanism to increase connectivity? Telescoping is an alternative to bone loss and fusion to increase connectivity?
379-384 “Due to sutures representing...” This sentence is also grammatically confusing. Also, does telescoping produce new sutures (meaning a new sutural ligament) or does it simply produce new contact between bones? In other words, does telescoping produce new contact without producing a new suture? Do the authors’ data or personal observations give insight to this?
406 I may have missed this, but what do the authors mean by “integration by proximity”? Are they juxtaposing this with integration by variation? This terminology should be clearly defined earlier in the manuscript.
458 What do the authors mean by important structural constraints? Also, could the authors expand upon and go into greater detail about what they mean by stronger functional and developmental codependence? This seems to be an interesting point they’re trying to make, but it needs to be fleshed out more.
467 What is meant by occipital-rostral bones here? Is this referring to the interdigitation between the maxilla and frontal, or something else?
520 Odontocete skulls were previously described as more complex, so this comment about slightly simpler skulls is confusing.

Tables/Figures
Table 1: Mark which network descriptors correspond with integration, complexity, etc. or provide this info in the caption
Figure 3: Label quadrants
Figure 4: The axis labels are illegible because they are too small; they need to be bigger. Also, it would be helpful if each plot had the same tick marks to facilitate comparison between plots. Archaeocetes appear to be missing from several plots, or aren’t clearly labeled.
Figure 5: A different style of arrow is needed in this figure. The arrows (> and <) too closely resemble less than or greater than symbols. This is particularly confusing, for example, when “less complex” is followed by the greater than symbol.
Supplemental Figures: Need a key for bone abbreviations.

Grammar/Typos
67 Word choice: the meaning of “experimented” is unclear here. Suggested replacements include “experienced” or “underwent” or “experimented with”
69 Remove “the” before telescoping
70 Typo: should be “extensive bone overlap”
67-70 Run-on sentence
73 horizontal sutures: Roston & Roth (2019) cite Gatesy et al. (2013) for this terminology, so Gatesy et al. should be cited here as well
87 Typo: “waipatiids”
93 Replace “Despite” with “while” or “although’ (e.g., “While telescoping was investigated in recent years...”)
99 Grammar: “studying” should be “study of”
102 “...has allowed its successful application...”
107 I was not sure what “This work” was referring to. At first, I thought the authors were referring to the present study, but the beginning of the next sentence suggests otherwise.
111 Typo: “archaeocete”
112 “In addition, we assessed if telescoping...”
122 Typo: archaeocetes
171 Typo: “...because they...”
193 “...toothed whales (Aetiocetus)...” This is a misleading way to write it, since Aetiocetus is a stem mysticete. Please rephrase
199 Typo: “...odontocete morphospace...”
224-225 Very confusing sentence.
279 Typo: “...remarkable cranial asymmetry...” Also, I recommend changing the word from development to evolution, as development is often interpreted as ontogeny.
292 typo: “especially”
306 typo: “overlap”
309 Typo. Remove “()”
328 missing period
330 “both reflecting” Should this say “also reflecting”? This sentence is a bit confusing.
356 typo: “dorsal exposure”
358-361 What kind of information and why would speed matter? The authors’ point is unclear here.
363 “...increment of modularity...”: Do the authors mean increased modularity?
370-372 “The amount of modules identified depends if the models of modularity identified a development correlation (in which case they identified 3 modules; del Castillo et al., 2017) or a functional correlation (between 5–10 modules; Churchill et al., 2018).” This sentence contains confusing grammar that obscures the authors’ intended meaning. The number of modules depends on whether the models are based on a developmental correlation or on a functional correlation?
387 Should not be plural: “Mysticete and odontocete skull network specializations”
393 Revise: “...the maxilla spreads over...”
399 revise: “...evolutionary framework, prograde telescoped skulls of mysticetes are...”
406 Revise: “...independent of...”
427 Might this also relate to the basicranial retroflexion of the odontocete skull?
430 Typo: “In addition to...”
437 Typo: “...compared with...”
443 Skulls should be plural here? Also, this sentence is missing a verb.
469 Typo: “...balaenopterid telescoping...”
477-479 The phrasing of this sentence is a bit confusing.
480-481 typo: “a stem odontocete”; Also, are they saying that Albertocetus has the smallest number of connections and least complex and least integrated skull?
477-487 Several typos: evidenced, exposure, platanistoid, network skull descriptors
492 I suggest a different word choice for patent, which can be confused with patent sutures (suture patency).
503-504 “The higher density...” This sentence appears to be missing a few words
523-526 Some confusing grammar & perhaps typos in these sentences.

---

## Round 0.2 · Minor Revisions

The revised manuscript has received a positive review from a new reviewer, Olivier Lambert, who has provided helpful, extensive, and detail-oriented edits (available as an annotated PDF). In the next round of revisions, I encourage the authors to heed Lambert's suggestions, as they will smooth out and clarify important parts of the manuscript. Equally, I ask that the authors pay close attention to word choice: given the preponderance of stem and crown concept usage, I don't see a need to say archaeocetes (or at least say stem cetaceans too); equally, I think "living" is a good substitute for "modern" which is scientifically colloquial and uninformative. Regarding sample size in odontocetes, que sera sera, and I don't see it as a show-stopper for this contribution.

Overall, I agree with Lambert's affirmative endorsement, and wish to contribute my own view: the revisions now address telescoping, modularity, and integration as separate but linked concepts; and the revised manuscript reads much better. On the whole, it is a substantial and important work that advances a century-old problem in cetacean biology in a meaningful way. PeerJ is lucky to serve as a venue for this work, which will be cited for years to come.

While the decision is minor revisions, we are now on the glide path towards acceptance. The requested changes are easy and minor, and the next revision will not need to go out to review.

·

Basic reporting

see below and previous review

Experimental design

see below and previous review

Validity of the findings

see below and previous review

Additional comments

Thank you for the detailed and convincing answers to many of the reviewers' concerns. The new version of the manuscript looks very fine. I only made a series of suggestions (in the marked .doc version of the main text) for generally very minor changes. In many cases, you should feel free to use alternate formulations.
I had to upload a pdf of the annotated text (with my changes and comments in red), but if it is easier I can also send the original .doc file by email.
I am looking forwards to seeing this very interesting work published (and maybe future analyses with a higher number of odontocete taxa sampled!).
O Lambert

---

## Round 0.3 · accepted · Accept

The authors' revisions are in line with all recommended edits and suggestions. This manuscript is now ready for acceptance!